# Significant Influence of a Single Atom Change in Auxiliary Acceptor on Photovoltaic Properties of Porphyrin-Based Dye-Sensitized Solar Cells

**DOI:** 10.3390/nano8121030

**Published:** 2018-12-11

**Authors:** Haoran Zhou, Jung-Min Ji, Min Su Kim, Hwan Kyu Kim

**Affiliations:** Global GET-Future Laboratory & Department of Advanced Materials Chemistry, Korea University, 2511 Sejong-ro, Sejong 339-700, Korea; zhouhaoran@naver.com (H.Z.); manbbong@korea.ac.kr (J.-M.J.); kimms38@korea.ac.kr (M.S.K.)

**Keywords:** D–π–A structural porphyrin, acceptor units, dye-sensitized solar cells, charge recombination, charge collection efficiency

## Abstract

The rational design of porphyrin sensitizers is always crucial for dye-sensitized solar cells (DSSCs), since the change of only a single atom can have a significant influence on the photovoltaic performance. We incorporated the pyridothiadiazole group, as a stronger electron-withdrawing group, into the commonly well-established skeleton of D-porphyrin-triple bond-acceptor sensitizers by a single atom change for a well-known strong electron-withdrawing benzothiadiazole (BTD) unit as an auxiliary acceptor. The impact of the pyridothiadiazole group on the optical; electrochemical; and photovoltaic properties of D–π–A porphyrin sensitizers was investigated with comparison for a benzothiadiazole-substituted **SGT-020** porphyrin. The pyridothiadiazole-substituted **SGT-024** porphyrin dye was red-shifted so that the absorption range might be expected to achieve higher light harvest efficiency (LHE) than the **SGT-020** porphyrin. However, all the devices were fabricated by utilizing **SGT-020** and **SGT-024**, evaluated and found to achieve a cell efficiency of 10.3% for **SGT-020**-based DSSC but 4.2% for **SGT-024**-based DSSC under standard global AM 1.5G solar light conditions. The main reason is the lower charge collection efficiency of **SGT-024**-based DSSC than **SGT-020**-based DSSC, which can be attributed to the tilted dye adsorption mode on the TiO_2_ photoanode. This may allow for faster charge recombination, which eventually leads to lower *J_sc_**, V_oc_* and power conversion efficiency (PCE).

## 1. Introduction

A huge amount of work has recently concentrated on third-generation solar cell technology development with low cost as emerging photovoltaics, such as dye-sensitized solar cells (DSSCs) [1,2], organic photovoltaics (OPVs) [3,4,5], perovskite solar cells (PSCs) [6,7,8], etc. Among the various solar technologies, DSSCs have garnered considerable attention due to the simple fabrication process, low cost, low toxicity, and high PCE under ambient lighting conditions [9,10]. In 1991 Grätzel and O’Regan first introduced mesoporous TiO_2_ nanocrystal layers into the DSSC system [11]. This led to a substantial improvement in photoelectric transformation efficiency. Since then, over the past 28 years, DSSCs have continued to show improved PCE [12,13,14]. Until now, state-of-the-art DSSCs have achieved PCEs approaching >11.9% for ruthenium complexes [15], >14% for metal-free D–π–A structural organic sensitizers [16], and 14.64% for D–π–A structural organic sensitizer-based tandem DSSCs under standard (1.5) illumination [13]. In comparison with ruthenium sensitizers [17,18] and metal-free sensitizers [19,20], D–π–A structural porphyrin sensitizers [21,22,23] have been attractive due to their extremely high molar extinction coefficient, exceptional ability to harvest light, and high photostability. To date, porphyrin-based DSSCs have achieved PCE by more than 13% under standard (1.5) illumination [24,25].

However, there are also some drawbacks due to the nature of porphyrins, such as the weak absorption in the range of 500–600 nm, the lack of absorption in the near-infrared region (NIR), and the dye aggregation caused by the extended π-conjugation structure [26,27,28,29,30]. It is apparent that these problems could be overcome by rational structural optimization. For example, long and well-adjusted alkoxy chains were introduced to porphyrin molecules, which significantly diminished the dye aggregation and reduced the interface back electron transfer rate [31]. The introduction of a benzothiadiazole (BTD) unit in the acceptor part to the well-established platform of d-porphyrin-triple bond-acceptor sensitizers was also a promising approach for elevating light-harvesting properties, as well as the photovoltaic performance. Up to now, the BTD unit was one of the most commonly used auxiliary electron acceptors in the well-known skeleton of d-porphyrin-triple bond-BTD-acceptor sensitizers for DSSCs [32,33,34].

In order to reduce the HOMO (highest occupied molecular orbital)–LUMO (lowest unoccupied molecular orbital) energy gap as well as extend the absorption range, a boosted electron-withdrawing pyridothiadiazole unit was introduced into the well-established platform of D-porphyrin-triple bond-BTD-acceptor sensitizers by a single atom change for the well-known strong electron-withdrawing benzothiadiazole (BTD) unit as an auxiliary acceptor. Although the pyridothiadiazole unit has been widely used in D–A polymers in OPVs [35,36] and metal-free organic sensitized solar cells [37,38], no application of the pyridothiadiazole unit has been explored in porphyrin-sensitized solar cells. Thus, based on our previously reported dye of **SGT-020 [25]**, we expected that the pyridothiadiazole unit in D–π–A porphyrin sensitizers could improve the absorption ability in NIR as well as the light harvest efficiency. Thus, a novel D–π–A structural porphyrin sensitizer was designed and synthesized, named as **SGT-024**, as shown in Scheme 1. Meanwhile, the optical properties, electrochemical properties, and photovoltaic performances were systemically investigated.

## 2. Synthetic Procedure

All reagents were purchased from AAlfa Aesar (Haverhill, MA, USA), TCI (Tokyo, Japan), and Sigma–Aldrich (St. Louis, MO, USA) unless stated otherwise. The synthesis routes of **SGT-024** were shown in Scheme 2, and compound **1** [37], compound **3** [25], and **SGT-020** [25] were synthesized according to the respective literature procedures. Details on the synthetic procedure, instrumentation, DSSC fabrication, photovoltaic parameters, ^1^H-NMR, ^13^C-NMR, and MALDI-TOF data are given in the Appendix A.

## 3. Results and Discussion

The UV-visible spectra of the dyes **SGT-020** and **SGT-024** in THF were collected and reported in Figure 1. It is obvious that **SGT-020** and **SGT-024** showed two intense absorption regions within the ranges of 400 to 500 nm (Soret band) and 600 to 800 nm (Q band). In Table 1, compared with the molar extinction coefficient (ε) of 143,040 M^−1^ cm^−1^ at the maximum absorption wavelength (λ_max_ = 454 nm) observed for **SGT-020**, the λ_max_ of **SGT-024** is 430 nm with a coefficient of 123,162 M^−1^ cm^−1^. It should be noted that the Soret band peaks of **SGT-024** were blue-shifted but the Q bands were significantly red-shifted when a pyridothiadiazole unit was introduced into the platform of D-porphyrin-triple bond-acceptor sensitizers by a single atom change for the well-known strong electron-withdrawing benzothiadiazole (BTD) unit. Furthermore, the fluorescent emission spectra were also measured in THF, as shown in Figure 1. **SGT-020** and **SGT-024** exhibited major emission bands at 724 and 791 nm, respectively. Therefore, the optical properties of these two porphyrin sensitizers revealed that the light capture region could be expanded by introducing the stronger electron-withdrawing moiety of the pyridothiadiazole unit.

The electrochemical properties of **SGT-020** and **SGT-024** were evaluated in THF with 0.1 M TBAPF_6_ as an electrolyte, using cyclic voltammetry (CV) (see Appendix A); the corresponding data are collected in Table 1. As shown in Figure 2, due to the fact that they have the same donor unit, their ground state oxidation potentials, which correspond to the HOMO energy levels of **SGT-020** and **SGT-024**, are nearly the same. Meanwhile, from the HOMO–LUMO band gap and the oxidation potential, the reduction potentials of **SGT-020** and **SGT-024** were determined to be −0.93 and −0.81 eV, respectively. This difference in reduction potentials could be ascribed to the structural changes in the acceptor group. The results also revealed that the reduction potential values are much more negative than the conducting band (CB) of TiO_2_ and the oxidation potential values are much more positive than the Co(bpy)_3_^2+/3+^ (bpy = 2,2′-bipyridine) redox couple (0.56 V vs. NHE), indicating that all of the electron transfer processes for two porphyrin sensitizers should occur efficiently due to the sufficient driving force.

DFT calculations at the M06 [39]/6-31G [40] (LANL2DZ [41] for Zn atom) level were carried out to better understand the electron distribution and the molecular geometries. The optimized ground state molecular structures and dihedral angles (between BTD/pyridothiadiazole unit and benzoic acid) of two porphyrins are shown in Figure 3 and Appendix A. The electron distribution of the HOMO energy levels was mainly delocalized at the diphenylamine donor unit and porphyrin core. However, as for the electron distribution of the LUMO energy levels, **SGT-024** showed a more evident shift to the unit of benzoic acid than **SGT-020**. Thus, **SGT-024** was expected to show enhanced intramolecular charge transfer (ICT) compared to that of **SGT-020**. Furthermore, in the optimized structures of **SGT-020** and **SGT-024**, the dihedral angles between the adjacent auxiliary acceptor and benzoic acid are 35.02° and 16.06°, respectively. Therefore, owing to the significant reduction of the torsion angle, **SGT-024** displayed more efficient π-conjugation and a better coplanar geometry in comparison with **SGT-020**, but this geometry would increase the possibility of dye aggregation.

The photovoltaic characteristics of the **SGT-020**- and **SGT-024**-based devices were measured under standard AM 1.5 conditions. The porphyrin-based TiO_2_ films were used as the photoanode in DSSCs, employing the Co(bpy)_3_^2+/3+^redox couple as electrolyte and CDCA (chenodeoxycholic acid) as co-adsorbent. The relevant photocurrent density‒voltage (J‒V) curves are shown in Figure 4a and the device parameters are summarized in Table 2. As compared to the reference dye **SGT-020** (*J*_sc_ = 14.8 mA cm^−2^, *V*_oc_ = 0.806 V, FF = 73.2% and PCE = 8.7%), the pyridothiadiazole-incorporated dye **SGT-024** only showed a moderate PCE of only 1.7%. The *J*_sc_ value of **SGT-024** dramatically dropped to 3.3 mA cm^−2^ and its *V*_oc_ also decreased to 0.655 V. To further investigate the *J_sc_* value of each porphyrin sensitizer, the corresponding IPCE (incident photon-to-electron conversion efficiency) spectra of **SGT-020** and **SGT-024**-based devices were measured under AM 1.5G solar light. As shown in Figure 4b, **SGT-024** sensitizers exhibited a broader, weaker absorption response (the absorption onset extended to almost 900 nm) than **SGT-020** (to ~850 nm), which displayed a similar tendency with the absorption spectra on TiO_2_ film, as shown in Appendix A. On the contrary, the **SGT-024**-based DSSCs exhibited an extremely low IPCE value (no more than 20%) from 400 to 900 nm; thus, lower *J*_sc_ values were observed.

In order to further improve the photovoltaic performance of **SGT-020** and **SGT-024**-based DSSCs, another co-adsorbent called **HC-A1** was introduced in this study. **HC-A1**, a multi-functional co-adsorbent widely used in our previous research [13,42], and its structure are shown in Appendix A. As expected, because of the light harvesting in shorter wavelength regions and efficient charge recombination retardation [43,44,45,46], the PCE of **SGT-020** and **SGT-024**-based solar cells was dramatically improved to 10.3% and 4.2%, respectively (see Figure 5). In addition, the dye loading amounts for **SGT-020** and **SGT-024** were found to be almost no different, implying that the effect of the dye loading amount on photovoltaic performance in this study is relatively small.

In order to get deeper insight into the photovoltaic performance difference as well as the interfacial charge transfer in all DSSC devices, the electrochemical impedance spectroscopy (EIS) was measured in the dark. Nyquist plots and Bode plots are shown in Figure 6 (a) and (b), respectively, and the EIS data are collected in Table 3. As far as we know, the first, second, and third semicircles correspond to the charge transfer resistance at the counter electrode, the resistance of TiO_2_/dye/electrolyte interface, and the diffusion resistance of Co(bpy)_3_^2+/3+^ redox couple in the electrolyte, respectively. The second semicircle of **SGT-024** (*R_rec_* = 6.12) was found to be much smaller than **SGT-020** (*R_rec_* = 16.8), indicating that the electron recombination rate of **SGT-024** is higher than that of **SGT-020**. On the other hand, according to the equation *τ*_r_ = *C*_µ_·*R*_rec_, the electron lifetime was calculated to be 5.12 or 3.33 ms for **SGT-020** and **SGT-024**, respectively. Meanwhile, the charge-collection efficiency *η*_cc_, derived from *η*_cc_ = (1 + *R*_tr_/*R*_rec_)^−1^ [47], was confirmed to be 84% for **SGT-020** and 68% for **SGT-024**. The results obtained above are consistent with the *V_oc_* values for the **SGT-020**-based device (0.795 V) and **SGT-024**-based device (0.724 V). Thus, when compared to **SGT-020** and **SGT-024**, the higher charge recombination rate and the lower charge collection efficiency of the **SGT-024**-based device may be the main reasons for its disappointing lower photovoltaic performance.

## 4. Conclusions

In this work, in order to investigate the structure‒performance relationship between the photovoltaic performance and the structure of various acceptors, the pyridothiadiazole group, as a stronger electron-withdrawing group, was incorporated into the well-established skeleton of D-porphyrin-triple bond-acceptor sensitizers by a single atom change for the well-known strong electron-withdrawing benzothiadiazole (BTD) unit. The impact of the pyridothiadiazole group on the optical, electrochemical, and photovoltaic properties of D–π–A porphyrin sensitizers was investigated by comparing with a benzothiadiazole-substituted **SGT-020** porphyrin. The porphyrin **SGT-024** presents a red-shifted and broadened Q-band in comparison with **SGT-020**, which could be attributed to the stronger electron-withdrawing nature of pyridothiadiazole than the **BTD** unit. This revealed that the introduction of pyridothiadiazole would be an effective strategy for strengthening the absorption of the well-established skeleton of D-porphyrin-triple bond-acceptor sensitizers, although it shows a more moderate PCE of 4.2% than the DSSC based on **SGT-020** (10.3%). The serious efficiency loss for the **SGT-024**-based device could be for two main reasons: the fast charge recombination rate caused by the strong electron-withdrawing acceptor, and thus lower charge collection efficiency observed; and the enhanced backbone co-planarity in **SGT-024** leading to unexpected dye aggregation. However, the pyridothiadiazole unit is still a promising synthetic strategy to explore D–π–A structural porphyrins with extended absorption properties. Our study has underlined the importance of a suitable auxiliary acceptor between the dye and the anchoring group for a sensitizer. This should be seriously considered in the further rational design of dye-sensitized solar cells.

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
