# Peer review of "Significant Influence of a Single Atom Change in Auxiliary Acceptor on Photovoltaic Properties of Porphyrin-Based Dye-Sensitized Solar Cells"

_nanomaterials, 2018, doi:10.3390/nano8121030_

Round 1

Reviewer 1 Report

The authors describe the synthesis of a novel dye molecule for us in dye-sensitized solar cells.  The dye is very similar to one that they previously synthesized and evaluated except for the substitution of a pyridothiadiazole group for a nearly identical group.  They then characterize the optical and electrochemical properties of the molecule and evaluate its use in DSSCs as compared to their previous work.  Although the new molecule is found to greatly underperform the original molecule, this knowledge would still be of interest to the DSSC community. 

Here are just a few areas that could use further comment or editing.

1.       Is there an explanation as to why the absorption and emission spectra of SGT-024 are broadened and red shifted in relation to SGT-020?

2.       It is stated correctly that the maximum extinction coefficient of SGT-020 is higher than SGT-024, but that is analyzing the light-harvesting efficiency at only one wavelength.  Is it possible to calculate a light harvesting efficiency (possibly in relation to the AM1.5 spectrum) over a range of wavelengths?

3.       In Figure 1, why is the emission data cut at 800 nm when it appears to extend beyond?  It would be beneficial to see the entire spectrum.

4.       The labels on the bars in Figure 2 read SGT-021 and SGT-023 and not SGT-020 and SGT-024 as stated in the caption.

5.       The section on charge recombination could use some clarification.  When you say that SGT-024 exhibits higher charge recombination rates, is this referring to recombination rate after charge transfer to TiO2 or excited state deactivation of the dye molecule?  I wouldn’t expect the rate to be any different once charge transfer has occurred, but a shorter excited state lifetime could lead to faster deactivation and a decreased efficiency of charge-carrier injection.  Time-dependent emission measurements could help distinguish between the two mechanisms.

Author Response

Please see an attached file.

Reviewer 2 Report

Zhou et al. present a comparative study between dye-sensitised solar cells employing two similarly structured porphyrin dyes, differentiated by a single atom. The effect of the substitution is substantial and a reminder of the fickle nature of such technologies. In general, this work has been performed well, although a few issues do need to be addressed:

* There is a mismatch between the short circuit current densities reported under simulated sunlight (14.8 and 3.3 mA/cm2) an those from integration of the IPCE. In the latter case I estimate values of 11.75 and 2.5 mA/cm2 respectively. This overestimation (around 25-30%) means the overall PCE is less impressive, but doesn't impact on the main story. See Nature Photonics volume 8, pages 669–672 (2014).

* Similar effects from minor changes in dye structure have been observed previously (eg. J. Mater. Chem. 22 pp7366-7379 (2012)), owing to changes made to the molecular orientation. Some discussion about such work should be presented in the introduction.

* While the term '3rd generation' PV has been applied by many authors to DSC and OPV, the distinction seems less scientific than the generations defined by Green et al. (ISBN-10 3-540-26562-7). I would also suggest to use a different abbreviation for perovskite solar cells as PrSC appears to suggest the use of Praseodymium.

* There is significant repetition of information between Fig 1a and 1b. I would suggest to either try to incorporate the information into one image, or move one (a?) to the supporting information. Is it possible to record the PL data slightly further into the red, as the tail is cut off here? Additionally, the axis label is partially covered for panel (a).

* In figure 2 the samples are names SGT-021 and SGT-023, whereas elsewhere (including the caption) they are SGT-020 and SGT-024.

* I suggest the authors use the word "should" instead of "would" on line 7 of page 5, is the relationship between chemical driving force and charge transfer is not always so straight forward.

* Measuring EIS in the dark at an arbitrary voltage is not necessarily indicative of real-world operational conditions. Also, i believe the interpretation may be backwards. Typically, the smaller the second semi-circle is the less resistance there is to charge transport through the TiO2 film, the better the device performance is. This would match better with their J-V responses.

* Axis scales on Nyquist plots should be the same for both directions (eg. 1 cm = 10 ohm/cm2). also note, these should be reported as ohm/cm2.

* The method for calculating charge collection efficiency mentioned here is very much a lower bound estimation and should be reported as such. In actuality, the charge collection efficiencies are likely to be much higher than these values. As a final point here, the authors should probably only report values such as these to whole percentage values (ie 84% and 68%).

Author Response

Please see an attached file.
